# What to Read in a Contract? Party-Specific Summarization of Legal Obligations, Entitlements, and Prohibitions

**Abhilasha Sancheti[†‡], Aparna Garimella[‡], Balaji Vasan Srinivasan[‡], Rachel Rudinger[†]**

[†]University of Maryland, College Park

[‡]Adobe Research

{sancheti, rudinger}@umd.edu

{sancheti, garimell, balsrini}@adobe.com

## Abstract

Reviewing and comprehending key obligations, entitlements, and prohibitions in legal contracts can be a tedious task due to their length and domain-specificity. Furthermore, the key rights and duties requiring review vary for each contracting party. In this work, we propose a new task of *party-specific* extractive summarization for legal contracts to facilitate faster reviewing and improved comprehension of rights and duties. To facilitate this, we curate a dataset comprising of *party-specific* pairwise importance comparisons annotated by legal experts, covering $\sim293K$ sentence pairs that include obligations, entitlements, and prohibitions extracted from lease agreements. Using this dataset, we train a pairwise importance ranker and propose a pipeline-based extractive summarization system that generates a *party-specific* contract summary. We establish the need for incorporating domain-specific notion of importance during summarization by comparing our system against various baselines using both automatic and human evaluation methods[1].

## 1 Introduction

A contract is a legally binding agreement that defines and governs the rights, duties, and responsibilities of all parties involved in it. To sign a contract (*e.g.*, lease agreements, terms of services, and privacy policies), it is important for these parties to precisely understand their rights and duties as described in the contract. However, understanding and reviewing contracts can be difficult and tedious due to their length and the complexity of legalese. Having an automated system that can provide an "at a glance" summary of rights and duties can be useful not only to the parties but also to legal professionals for reviewing contracts. While existing works generate section-wise summaries of unilateral contracts, such as terms of services (Manor and Li, 2019) or employment agreements (Balachandar

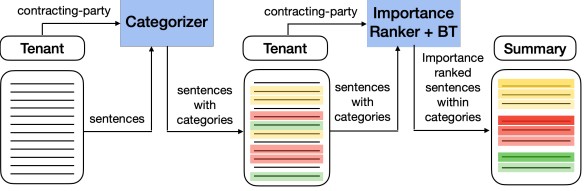

Figure 1: An illustration of input contract with a party (Tenant) and output from CONTRASUM. It first identifies sentences in a contract that contains any of the obligations , entitlements , and prohibitions specific to the party. Then, ranks the identified sentences based on their importance level (darker the shade, higher the importance) to produce a summary. BT: Bradley-Terry.

et al., 2021), as well as risk-focused summaries of privacy policies (Keymanesh et al., 2020), they aim to generate a single summary for a contract and do not focus on key rights and duties.

However, we argue that a single summary may not serve all the parties as they may have different rights and duties (Ash et al., 2020; Sancheti et al., 2022) (*e.g.,* a *tenant* is obligated to timely pay the rent to the landlord whereas, the *landlord* must ensure the safety and maintenance of rented property). Further, each right or duty may not be equally important (*e.g.*, an obligation to pay rent on time (otherwise pay a penalty) is more important for tenant than maintaining a clean and organized premise) and this importance also varies for each party. To address this, we introduce a new task of *party-specific* extractive summarization of *important* rights and duties in a contract. Motivated by the different categories in the software license summaries available at TL;DRLegal[2] describing what users *must*, *can*, and *cannot* do under a license, we define rights and duties in terms of deontic modalities (Matulewska, 2010; Peters and Wyner, 2016; Sancheti et al., 2022): *obligations* ("must"), *entitlements* ("can"), and *prohibitions* ("cannot"). In this work, we refer to these as (modal) categories.

---

[1]Data available at http://bit.ly/legal-importance

[2]https://tldrlegal.com/

Existing summarization systems perform poorly on legal contracts due to large compression ratios and the unavailability of public datasets (Manor and Li, 2019) that are large enough to train neural summarization systems (See et al., 2017; Paulus et al., 2017). While previous studies have either devised unsupervised methods to identify obligations, entitlements, and prohibitions for a party in a contract (Ash et al., 2020) or collected annotations for the same (Sancheti et al., 2022), they do not consider the *relative importance* of instances within each category. Although existing unsupervised summarization methods rely on sentence centrality-based importance measures (Mihalcea and Tarau, 2004; Erkan and Radev, 2004), capturing sentential meaning (Zheng and Lapata, 2019), or analyzing the relationship between sentences to capture the structure of document (Ozsoy et al., 2011), these methods fail to capture the legal-domain-specific notion of *importance* among sentences. To address these issues, we construct a legal-expert annotated dataset of *party-specific* pairwise importance comparisons between sentences from lease agreements. We focus on lease agreements, which have clearly defined parties with asymmetric roles (*tenant*, *landlord*), and for which modal category labels have previously been collected (Sancheti et al., 2022).

We break down the contract-level task of *party-specific* summarization into two sentence-level subtasks: (1) **Content Categorization** – identifying the modal categories expressed in the sentences of a contract for a specified party, and (2) **Importance Ranking** – ranking the sentences based on their importance for a specified party. This approach has three benefits: (a) enabling us to use an existing corpus for identifying deontic modalities in contract, (b) cognitively simplifying the annotation task for experts who only need to compare a few sentences (resulting in higher agreement) at a time rather than read and summarize a full contract (spanning $10 - 100$ pages), and (c) reducing the cost of annotation as a contract-level end-to-end summarization system requires more data to train than does a sentence-level categorizer and ranker.

This work makes the following **contributions**: (a) we introduce a new task of *party-specific* extractive summarization of *important obligations*, *entitlements*, and *prohibitions* in legal contracts; (b) we are the first to curate a novel legal expert annotated dataset (§4) (using best-worst scaling) consisting of *party-specific* pairwise importance comparisons

for sentence pairs (that include obligations, entitlements, or prohibitions) from lease agreements; and (c) we train a pairwise importance ranker using the curated dataset to build a pipeline-based extractive summarization system (§5) for the proposed task, and show the effectiveness of the system as compared to several unsupervised ranking-based summarization baselines, under both automatic and human evaluations (§8); underscoring the domain-sensitive nature of "importance" in legal settings.

## 2 Related Work

**Summarization of Legal Text** Existing works focus on summarizing legal case reports (Galgani and Hoffmann, 2010; Galgani et al., 2012; Bhattacharya et al., 2021; Agarwal et al., 2022; Elaraby and Litman, 2022), civil rights (Shen et al., 2022), terms of service agreements (Manor and Li, 2019), privacy policies (Tesfay et al., 2018; Zaeem et al., 2018; Keymanesh et al., 2020). Hybrid approaches combining different summarization methods (Galgani et al., 2012) or word frequency (Polsley et al., 2016) with domain-specific knowledge have been used for legal case summarization task. Most similar works either propose to identify sections of privacy policies with a high privacy risk factor to include in the summary by selecting the riskiest content (Keymanesh et al., 2020) or directly generate section-wise summaries for employment agreements (Balachandar et al., 2021) using existing extractive summarization methods (Kullback and Leibler, 1951; Mihalcea and Tarau, 2004; Erkan and Radev, 2004; Ozsoy et al., 2011). We differ from these works in that instead of generating a single summary, we generate *party-specific* summaries of key rights and duties for contracts by learning to rank sentences belonging to each of these categories based on their importance as decided by legal experts.

**Rights and Duties Extraction** Existing works either propose rule-based methods (Wyner and Peters, 2011; Peters and Wyner, 2016; Matulewska, 2010), or use a combination of NLP approaches such as syntax and dependency parsing (Dragoni et al., 2016) for extracting rights and obligations from legal documents such as Federal code regulations. Others (Bracewell et al., 2014; Neill et al., 2017; Chalkidis et al., 2018) use machine learning and deep learning approaches to predict rights and duties with the help of small datasets which are not publicly available. While rule-based unsupervised

approaches exist (Ash et al., 2020) to identify rights and duties with respect to a party, they are not flexible and robust to lexical or syntactic variations in the input. Funaki et al. (2020) curate an annotated corpus of contracts for recognizing rights and duties using LegalRuleML (Athan et al., 2013) but it is not publicly available. Sancheti et al. (2022) introduced a dataset of lease contracts with annotations for party-specific deontic categories and corresponding triggers mentioned in a sentence. We use the proposed model as the content categorizer.

## 3  Task Definition

We formally define the new task as: given a contract C consisting of a sequence of sentences ($c_1$, $c_2$, ..., $c_n$) and a party $P$, the task is to generate an extractive summary S consisting of the most important $m_o$ obligations, $m_e$ entitlements, and $m_p$ prohibitions (where $m_i < n \; \forall i \in \{o, e, p\}$) for the specified party. As previously mentioned, we break this task into two sub-tasks: (1) **Content Categorization** to identify the categories expressed in a sentence of a contract for a given party, and (2) **Importance Ranking** to rank the sentences based on their *importance* to a given party.

## 4  Dataset Curation

The LEXDEMOD dataset (Sancheti et al., 2022) contains sentence-level, party-specific annotations for the categories of *obligations*, *entitlements*, *prohibitions*, *permissions*, *non-obligations*, and *non-entitlements* in lease agreements. However, the data does not contain any importance annotations. To facilitate importance ranking, we collect *party-specific* pairwise importance comparison annotations for sentences in this dataset. We only consider the three major categories: *obligations*, *entitlements*[3], and *prohibitions*.

**Dataset Source**   We use a subset of contracts from the LEXDEMOD dataset to collect importance annotations as it enables us to create reference summaries for which we need both the category and importance annotations. LEXDEMOD contains lease agreements crawled from Electronic Data Gathering, Analysis, and Retrieval (EDGAR) system which is maintained by the U.S. Securities and Exchange Commission (SEC). The documents filed

on SEC are public information and can be redistributed without further consent[4].

**Annotation Task**   Rating the *party-specific* importance of a sentence in a contract on an absolute scale requires well-defined importance levels to obtain reliable annotations. However, defining each importance level can be subjective and restrictive. Moreover, rating scales are also prone to difficulty in maintaining inter- and intra-annotator consistency (Kiritchenko and Mohammad, 2017), which we observed in pilot annotation experiments. We ran pilots for obtaining importance annotations for each sentence in a contract, as well as a pair of sentences on a scale of 0-5 taking inspiration from prior works (Sakaguchi et al., 2014; Sakaguchi and Van Durme, 2018) but found they had a poor agreement (see details in A.1). Thus, following Abdalla et al. (2021), we use Best–Worst Scaling (BWS), a comparative annotation schema, which builds on pairwise comparisons and does not require $\binom{N}{2}$ labels. Annotators are presented with $n=4$ sentences from a contract and a party, and are instructed to choose the best (*i.e.*, most important) and worst (*i.e.*, least important) sentence. Given $N$ sentences, reliable scores are obtainable from ($1.5N$–$2N$) 4-tuples (Louviere and Woodworth, 1991; Kiritchenko and Mohammad, 2017). To our knowledge, this is the first work to apply BWS for importance annotation in legal contracts.

From a list of $N = 3,300$ sentences spanning 11 lease agreements from LEXDEMOD, we generate[5] $4,950$ ($1.5N$) unique 4-tuples (each consisting of 4 distinct sentences from an agreement) such that each sentence occurs in at least six 4-tuples. We hire 3 legal experts (lawyers) from Upwork with $>2$ years of experience in contract drafting and reviewing to perform the task. We do not provide a detailed technical definition for importance but brief them about the task of summarization from the perspective of review and compliance, and encourage them to rely on their intuition, experience, and expertise (see below section for annotation reliability). Each tuple is annotated by 2 experts.

**Annotation Aggregation**   Annotation for each 4-tuple provides us 5 pairwise inequalities. *E.g.*, if $a$ is marked as the most important and $d$ as the least important, then we know that $a \succ b$, $a \succ c$,

---

[3]We merge *entitlement* and *permission* categories as there were a small number of permission annotations per contract. Entitlement is defined as a right to have/do something while permission refers to being allowed to have/do something

[4]https://www.sec.gov/privacy.htm#dissemination
[5]The tuples are generated using the BWS scripts provided in Kiritchenko and Mohammad (2017): http://saifmohammad.com/WebPages/BestWorst.html

| Party | %Obl.≻Ent. | %Ent.≻Pro. | %Pro.≻Obl. |
|-------|-----------|-----------|-----------|
| **Tenant** | **52.25** | 39.31 | **54.49** |
| **Landlord** | 46.56 | **55.98** | 46.82 |

Table 1: Importance comparison statistics (per party) computed from a subset of pairwise comparisons where the category of the sentences in a pair is different. Bold win %s are significant ($p < 0.05$). ≻: more important.

$a \succ d$, $b \succ d$, and $c \succ d$. From all the annotations and their inequalities, we calculate real-valued importance scores for all the sentences using Bradley-Terry (BT) model (Bradley and Terry, 1952) and use these scores to obtain $\binom{N}{2}$ pairwise importance comparisons. BT is a statistical technique used to convert a set of paired comparisons (need not be a full or consistent set) into a full ranking.

**Annotation Reliability**   A commonly used measure of quality and reliability for annotations producing real-valued scores is split-half reliability (SHR) (Kuder and Richardson, 1937; Cronbach, 1951). It measures the degree to which repeating the annotations would lead to similar relative rankings of the sentences. To measure SHR[6], annotations for each 4-tuple are split into two bins. These bins are used to produce two different independent importance scores using a simple counting mechanism (Orme, 2009; Flynn and Marley, 2014): the fraction of times a sentence was chosen as the most important minus the fraction of times the sentence was chosen as the least important. Next, the Spearman correlation is computed between the two sets of scores to measure the closeness of the two rankings. If the annotations are reliable, then there should be a high correlation. This process is repeated 100 times and the correlation scores are averaged. Our dataset obtains a SHR of $0.66 \pm 0.01$ indicating moderate-high reliability.

**Dataset Analysis**   We obtain a total of $293,368$ paired comparisons after applying the BT model. Table 1 shows the percentage of sentences containing a category annotated as more important than those from another. For tenants, prohibitions are annotated as more important than both obligations and entitlements, perhaps as prohibitions might involve penalties. Similarly, obligations are more important than entitlements, as knowing them might be more important for tenants (*e.g.,* failure to

---

[6]We use scripts available at https://www.saifmohammad.com/WebDocs/Best-Worst-Scaling-Scripts.zip.

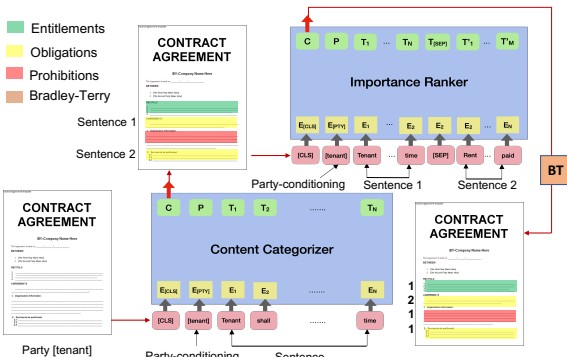

Figure 2: CONTRASUM takes a contract and a party to first identify all the sentences containing party-specific obligations, entitlements, and prohibitions using a content categorizer. Then, the identified sentences for each category are pairwise-ranked using an importance ranker. A ranked list of sentences is obtained using the Bradley-Terry model to obtain the final summary.

perform duties such as "paying rent on time" may lead to late fees). For landlords, on the other hand, knowing entitlements is more important than obligations or prohibitions, possibly due to the nature of lease agreements where landlords face fewer prohibitions and obligations than tenants.

As we do not explicitly define the notion of "importance" during annotation. We learned from feedback and interaction with the annotators about several factors they considered when determining importance. These factors included the degree of liability involved (*e.g.,* 'blanket indemnifications' were scored higher than 'costs incurred in alterations' by a tenant since blanket rights can have an uncapped amount of liability), and liabilities incurred by a more probable event were rated as more important than those caused by less probable events, among others. As is evident from these examples, the factors that can influence importance may be complex, multi-faceted, and difficult to exhaustively identify, which is why our approach in this work was to allow our annotators to use their legal knowledge to inform a holistic judgment.

## 5   CONTRASUM: Methodology

We build a pipeline-based extractive summarization system, CONTRASUM (Figure 2), for the proposed task. It consists of two modules for the two sub-tasks: **Content Categorizer** (§5.1) and **Importance Ranker** (§5.2). The Content Categorizer first identifies sentences from a contract that mention any of the obligations, entitlements, or prohibitions for a given party. Then, the identified sentences per

category for a given party are pairwise ranked for importance using the Importance Ranker. From all the pairwise comparisons, a ranked list of sentences is obtained using the Bradley-Terry model (Bradley and Terry, 1952) to produce a final party-specific summary consisting of most important $m_o$ obligations, $m_e$ entitlements, and $m_p$ prohibitions.

## 5.1 Content Categorizer

The Content Categorizer takes in a sentence $c_i$ from C and a party to output all the categories (such as, *obligations, entitlements, prohibitions*) mentioned in $c_i$. Such categorization helps in partitioning the final summary as per the categories of interest. We use the multi-label classifier introduced in (Sancheti et al., 2022) as the categorizer. Since a sentence can contain multiple categories, the same sentence can be a part of different categories.

## 5.2 Importance Ranker

A contract may contain a large number of obligations, entitlements, and prohibitions for each party; however, not all instances of each category may be equally important. Also, the importance of sentences within each category may vary for each party. Therefore, this module aims to rank the sentences belonging to each category based on their level of *importance* for the specified party. As indicated in §4, we do not define the notion of "importance"; instead, we rely on the annotations from legal experts based on their understanding from contract review and compliance perspective to build the ranker. This module (Figure 2) takes in a pair of sentences $(c_i, c_j)$ from C and a party $P$ to predict if $c_i$ is more important to be a part of the summary than $c_j$ for $P$. We model this as a binary classifier that orders the input pair of sentences based on their importance for the specified party.

## 5.3 End-to-end Summarization

Recall that the Content Categorizer and Importance Ranker work at the sentence- and sentence-pair-level, respectively. Therefore, to produce the desired *party-specific* category-based summary for a contract, we obtain (1) categories for each sentence in the contract (using the Content Categorizer), and (2) a ranking of the sentence pairs within each category according to their relative importance (using the Importance Ranker) with respect to the party. We do not explicitly account for the diversity within each category (although organization by category

| Label/Split | Train | Dev | Test |
|---|---|---|---|
| Positive | 85529 | 2538 | 71576 |
| Negative | 62857 | 2137 | 68731 |

Table 2: Statistics of pairwise importance annotations.

helps ensure some degree of diversity across categories). As the ranker provides ranking at a sentence pair level, to obtain an importance-ranked list of sentences from all the pairwise predictions for each category, we use the Bradley-Terry (BT) model as described in §4. We produce the final summary by selecting the $m_o$, $m_e$, and $m_p$ most important sentences predicted as obligations, entitlements, and prohibitions, respectively.

## 6 Experimental Setup

CONTRASUM is a pipeline-based system consisting of two modules. The two modules are trained (and evaluated) separately and pipelined for generating end-to-end summaries as described in §5.3.

**Datasets**  We use the category annotations in LEXDEMOD dataset (Sancheti et al., 2022) to train and evaluate the **Content Categorizer**. Dataset statistics are shown in Table 10 (§A.2).

For training the **Importance Ranker**, we need sentence pairs from contracts ordered by relative importance to a particular party. We use the pairwise importance comparison annotations collected in §4 to create a training dataset. If, for a pair of sentences (a,b), $a \succ b$ (a is more important than b), then the label for binary classification is positive, and negative otherwise. We retain the same train/dev/test splits as LEXDEMOD to avoid any data leakage. Table 2 present the data statistics.

As mentioned earlier, the two modules are separately trained and evaluated at a sentence- or sentence pair-level. However, CONTRASUM generates *party-specific* summaries at a contract-level. Therefore, for evaluating the system for the end-to-end summarization task, we need reference summaries. To obtain reference summaries, we need ground-truth category and importance ranking between sentences in a contract. Since we collected importance comparison annotations for the same sentences for which LEXDEMOD provides category annotations, we group the sentences belonging to a category (ground-truth) and then derive the ranking among sentences within a category using the gold importance annotations and BT model (as

described earlier) for each party. We obtain reference summaries at different compression ratios (CR) $(5\%, 10\%, 15\%)$ with a maximum number of sentences capped at 10 per category to evaluate the output summaries against different reference summaries. CR is defined as the % of total sentences included in the summary for a contract. Please note that the reference summaries are extractive.

**Training Details** We use the RoBERTa-large (Liu et al., 2019) model with a binary (multi-label) classification head and fine-tune it on the collected importance dataset (LEXDEMOD) dataset for the **Importance Ranker** (**Content Categorizer**) using binary-cross entropy loss.

**Implementation Details** We use HuggingFace's Transformers library (Wolf et al., 2019) to fine-tune PLM-based classifiers. The content categorizer is fine-tuned for 20 epochs, and the importance ranker for 1 epoch with gradient accumulation steps of size 4. We report the test set results for model(s) with the best F1-macro score on the development set. Both the category and importance classifiers are trained with maximum sequence length of 256 and batch size of 8. For both the classifiers, party conditioning is done with the help of special tokens prepended to the sentence, as this has been successfully used previously for controlled text generation tasks (Sennrich et al., 2016; Johnson et al., 2017). The input format for the Content Categorizer is - [PARTY] SENTENCE and for the Importance Ranker is - [PARTY] SENTENCE1 $\langle/s\rangle$ SENTENCE2. More details are provided in A.6. **Dataset Pre-processing.** We use lexnlp (Bommarito II et al., 2021) to get sentences from a contract and filter the sentences that define any terms (using regular expressions such as, "means", "mean", and "shall mean") or does not mention any of the parties (mentioned in the form of alias such as, "Tenant", "Landlord", "Lessor", and "Lessee").

## 7 Evaluation

We perform an extrinsic evaluation of CONTRA-SUM to assess the quality of the generated summaries as well as an intrinsic evaluation of the two modules separately to assess the quality of categorization and ranking. Intrinsic evaluation of the Categorizer and the Ranker is done using the test sets of the datasets (§6) used to train the models. Although the size of datasets used to evaluate the two modules at a sentence- ($\sim1.5K$ for Categorizer) or

sentence pair-level ($\sim130K$ for Ranker) is large, it amounts to 5 contracts for the extrinsic evaluation of CONTRASUM for the end-to-end summarization task at a contract-level. Since the number of contracts is small, we perform 3-fold validation of CONTRASUM. Each fold contains 5 contracts (in the test set) sampled from 11 contracts (remaining contracts are used for training the modules for each fold) for which both category and importance annotations are available to ensure the availability of reference summaries. CONTRASUM uses the best Categorizer and the best Ranker (RoBERTa-L models) trained on each fold (see Table 3 and 7 for results on fold-1;A.5 for other folds).

**Evaluation Measures** We report macro-averaged Precision, Recall, and F1 scores for predicting the correct categories or importance order for the **Content Categorizer** and **Importance Ranker**. We also report the accuracy of the predicted labels. As both the reference and predicted summaries are extractive in nature, we use metrics from information retrieval and ranking literature for the end-to-end evaluation of CONTRASUM. We report Precision@k, Recall@k, F1@k (following Keymanesh et al., 2020), Mean Average Precision (MAP), and Normalized Discounted Cumulative Gain (NDCG) (details on formulae in A.3) computed at a category-level for each party; these scores are averaged across categories for each party and then across parties for a contract. We believe these metrics give a more direct measure of whether any sentence is correctly selected and ranked in the final summary. For completeness, we also report ROUGE-1/2/L scores in Table 9 in A.5.

**Importance Ranker Baselines** We compare the Ranker against various pre-trained language models; **BERT-BU** (Devlin et al., 2018), contract BERT (**C-BERT-BU**) (Chalkidis et al., 2020), and **RoBERTa-B** (Liu et al., 2019), and majority.

**Content Categorizer Baselines** We do not compare the categorizer against other baselines, instead directly report the scores in Table 7, as we use the model with the best F1-score on the development set as described in (Sancheti et al., 2022).

**Extractive Summarization Baselines** We compare CONTRASUM against several unsupervised baselines which use the same content categorizer but the following rankers.

| Model | Accuracy | Precision | Recall | F1 |
|---|---|---|---|---|
| Majority | 51.58/50.35/51.01 | 25.79/25.17/25.51 | 50.00/50.00/50.00 | 34.03/33.49/33.78 |
| BERT-BU | 69.80/67.72/68.83 | 69.77/67.73/68.81 | 69.75/67.71/68.80 | 69.76/67.71/68.80 |
| RoBERTa-B | 70.50/68.11/**69.75** | 70.47/68.15/68.52 | 70.46/68.09/68.43 | 70.46/68.08/68.41 |
| C-BERT-BU | 70.32/68.24/69.29 | 70.30/68.25/68.94 | 70.28/68.23/69.06 | 70.29/68.22/68.97 |
| RoBERTa-L | **70.55/68.54**/69.60 | **70.53/68.61/69.62** | 70.47/**68.51/69.54** | **70.48/68.49/69.54** |

Table 3: Evaluation results for the **Importance Ranker**. Scores for **Tenant/Landlord/Both** are averaged over 3 different seeds. BU, B, and L denote base-uncased, base, and large, respectively.

1. **Random** baseline picks random sentences from the set of predicted sentences belonging to each category. We report average scores over 5 seeds.
2. **KL-Sum** (Kullback and Leibler, 1951) aims to minimize the KL-divergence between the input contract and the produced summary by greedily selecting the sentences.
3. **LSA** (Latent Semantic Analysis) (Ozsoy et al., 2011) analyzes the relationship between document sentences by constructing a document-term matrix and performing singular value decomposition to reduce the number of sentences while capturing the structure of the document.
4. **TextRank** (Mihalcea and Tarau, 2004) uses the PageRank algorithm to compute an importance score for each sentence. High-scoring sentences are then extracted to build a summary.
5. **LexRank** (Erkan and Radev, 2004) is similar to TextRank as it models the document as a graph using sentences as its nodes. Unlike TextRank, where all weights are assumed as unit weights, LexRank utilizes the degrees of similarities between words and phrases, then calculates an importance score for each sentence.
6. **PACSUM** (Zheng and Lapata, 2019) is another graph-based ranking algorithm that employs BERT (Devlin et al., 2018) to better capture the sentential meaning and builds graphs with directed edges as the contribution of any two nodes to their respective centrality is influenced by their relative position in a document.
7. **Upper-bound** picks sentences from a contract for a category based on the ground-truth importance ranking derived from human annotators. It indicates the performance upper-bound of an extractive method that uses our categorizer.

We limit the summaries to contain $m_i$=10 (chosen subjectively) sentences per category for experimental simplicity and that it is short enough to be processed quickly by a user while capturing the most critical points but this number can be adjusted. Note that if less than $m_i$ sentences are predicted to belong to a category then the summary will have

| | Model | CR=0.05 | | CR=0.10 | | CR=0.15 | |
|---|---|---|---|---|---|---|---|
| | | MAP | NDCG | MAP | NDCG | MAP | NDCG |
| PC | Random | 0.113 | 0.169 | 0.123 | 0.199 | 0.141 | 0.228 |
| | KL-Sum | 0.126 | 0.173 | 0.132 | 0.194 | 0.138 | 0.218 |
| | LSA | 0.093 | 0.154 | 0.103 | 0.177 | 0.142 | 0.225 |
| | TextRank | 0.121 | 0.182 | 0.133 | 0.219 | 0.152 | 0.244 |
| | PACSUM (BERT) | 0.135 | 0.194 | 0.148 | 0.232 | 0.150 | 0.249 |
| | LexRank | 0.130 | 0.186 | 0.151 | 0.226 | 0.165 | 0.253 |
| | CONTRASUM | **0.223** | **0.319** | **0.234** | **0.351** | **0.262** | **0.392** |
| PC | Upper-bound | 0.579 | 0.628 | 0.607 | 0.680 | 0.614 | 0.698 |
| GC | Random | 0.206 | 0.298 | 0.209 | 0.316 | 0.228 | 0.346 |
| | KL-Sum | 0.174 | 0.252 | 0.191 | 0.293 | 0.204 | 0.313 |
| | LSA | 0.234 | 0.309 | 0.239 | 0.332 | 0.287 | 0.389 |
| | TextRank | 0.144 | 0.237 | 0.151 | 0.264 | 0.178 | 0.303 |
| | PACSUM (BERT) | 0.162 | 0.254 | 0.178 | 0.291 | 0.210 | 0.330 |
| | LexRank | 0.198 | 0.278 | 0.219 | 0.317 | 0.236 | 0.353 |
| | CONTRASUM-CC | 0.398 | 0.525 | 0.393 | 0.535 | 0.431 | 0.575 |

Table 4: Evaluation results of end-to-end summarization for different compression ratios (CR) averaged across 3 folds. GC (PC) denotes use of ground-truth (predicted from CC) categories. CC: Content Categorizer.

less than $m_i$ sentences. We also compare the performance of summarization systems that utilize the ground-truth categories (GC) followed by the above ranking methods to produce the summary. These systems help us investigate the effect of error propagation from the categorizer to the ranker and the final summaries. We use the Sumy[7] python package for implementing these ranking methods.

## 8 Results and Analysis

**Automatic Evaluation of the Categorizer and the Ranker** We report the evaluation results for the Ranker in Table 3. Fine-tuned PLMs outperform the majority baseline on F1 score as expected. While C-BERT-BU, which is pre-trained on contracts, performs better than BERT-BU and RoBERTa-B, the overall F1 score is the highest for RoBERTa-L, suggesting increased model size and training compensate for lack of pretraining on contracts. For the Categorizer, we report results from (Sancheti et al., 2022) as-is in Table 7 in A.5.

**Automatic Evaluation of Summaries** We report the automatic evaluation results for the end-to-end summarization of contracts in Table 4. CONTRASUM achieves the best scores for both MAP and

---
[7] https://pypi.org/project/sumy/

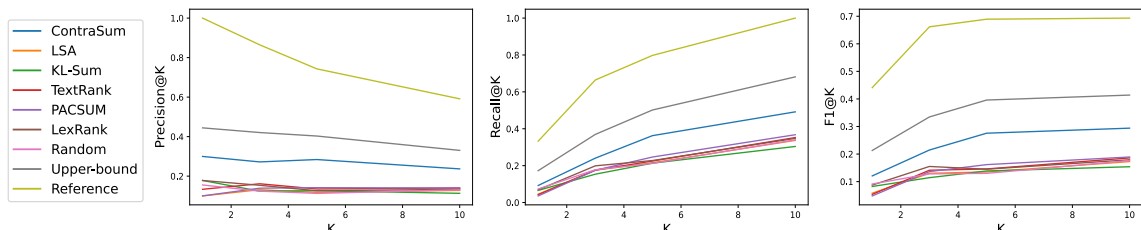

Figure 3: Precision@K, Recall@K, and F1@K ($K \in \{1, 3, 5, 10\}$) scores averaged across 3 folds for the end-to-end summaries at CR=0.15. All the models (except Reference) use predicted categories for the final summary.

NDCG computed against the gold-references at different compression ratios establishing the need for domain-specific notion of importance which is not captured in the other baselines. Surprisingly, Random baseline performs better than (LSA) or is comparable to other baselines (KL-Sum) when predicted categories are used. While PACSUM achieves better scores than LexRank at low compression ratios, using centrality does not help at CR=0.15. As expected, we observe a consistent increase in NDCG with the increase in the compression ratio as the number of sentences per category increases in the gold-references. While CONTRA-SUM outperforms all the baselines, it is still far away from the upper-bound performance which uses the gold importance ranking. This calls for a more sophisticated and knowledge-driven approach to learning the importance of different sentences.

As CONTRASUM is a pipeline-based system where the sentences predicted as containing each of the categories are input to the importance ranker, erroneous category predictions may affect the final summaries. Thus, we present scores (last block in Table 4) from different systems that use ground-truth (GC) categories and various ranking methods. While the performance of all the systems improves over using predicted categories, the random baseline with GC outperforms all other baselines with GC, except for LSA, suggesting off-the-shelf summarizers are not well-suited for this task. Nevertheless, CONTRASUM beats all the systems without the Categorizer indicating the effectiveness of the Importance Ranker in comparing the importance of sentences. We report party-wise results in A.5 (Table 11) and an output summary in Figure 4 and 5.

Figure 3 shows the Precision@k, Recall@k, and F1@k$\in \{1, 3, 5, 10\}$ trends. As expected, precision decreases with $k$ while recall and F1 increase. Similar to Table 4, CONTRASUM outperforms the baselines in each plot while there is a huge gap between our model's performance and the upper-

| Model | Info.↑ | Use↑ | AoC↑ | Red.↓ | AoIR↑ | O↑ |
|-------|--------|------|------|-------|-------|------|
| Rand | 2.96 | 2.87 | 2.92 | 1.92 | 2.50 | 2.94 |
| LR | 3.00 | 2.79 | 3.04 | 2.04 | 2.46 | 3.06 |
| CS | **3.25** | **3.37** | **3.25** | 2.04 | 2.54 | **3.50** |
| Ref | 3.17 | 3.00 | 3.17 | **1.75** | **2.87** | 3.06 |

Table 5: Results for human evaluation of model summaries. Mean scores of the 2 annotators are averaged across categories and parties. Overall (O) rating is scaled down to half to make it comparable. Info., Use, AoC, Red., AoIR denote informativeness, usefulness, accuracy of categorization, redundancy, and accuracy of importance ranking, resp. Rand: Random (PC), LR: LexRank (PC), CS: CONTRASUM, Ref: References.

bound leaving scope for improvement in the future.

Owing to the recent advancements and power of LLMs, we prompt ChatGPT (details in §A.5) to asses its performance on this task. We find that it is not straightforward for ChatGPT to perform the task with simple prompting due to hallucinations in the generated output and token limit. Further work is needed to look into how to best use LLMs for such tasks in domains such as legal.

**Human Evaluation of Summaries** In addition to automatic evaluation, we also give contracts along with their *party-specific* summaries from CONTRASUM, gold-references (CR=0.15), the best baseline-LexRank (PC), and Random (PC) baseline to legal experts for human evaluation. 16 summaries, each consisting of a maximum of 10 sentences per category, for 2 contracts are provided to 2 experts. They are asked to rate the summaries for each category on a 5-point scale (1 least; 5 most) as per: (1) informativeness; (2) usefulness; (3) accuracy of categorization; (4) redundancy, and (5) accuracy of importance ranking. In addition, we ask them to rate the overall quality of a summary on a 10-point scale. The average scores are presented in Table 5. CONTRASUM produces the best summaries overall but lacks in the diversity of sentences within each category. Interestingly,

experts found summaries from CONTRASUM to be more informative, useful, and correctly categorized than the gold-references. This may happen as both predicted and reference summaries are capped at 10 sentences per category however, because the category labels themselves were predicted, the system and reference summaries did not always contain the same number of sentences per category if one was below the allowed limit (and in principle it is not possible to enforce a minimum number of sentences per category). It is surprising that possible miscategorizations led to higher human ratings of overall summary outputs; this is an unexpected finding that highlights the potential challenges of evaluating complex, structured summarization outputs such as in this task (see §A.4 for detailed discussion). Furthermore, experts also considered the importance of a category expressed in a sentence to a party. Hence, they penalize for the accuracy of categorization if the category is indirectly (although correctly) expressed in a sentence for a party (see B for an example, category-wise evaluation results, and more details).

## 9 Conclusions

We introduced a new task to extract *party-specific* summaries of *important obligations*, *entitlements*, and *prohibitions* in legal contracts. Obtaining absolute importance scores for contract sentences can be particularly challenging, as we noted in pilot studies, thus indicating the difficulty of this task. Instead, we collected a novel dataset of legal expert-annotated *pairwise importance comparisons* for $293K$ sentence pairs from lease agreements to guide the Importance Ranker of CONTRASUM built for the task. Automatic and human evaluations showed that our system that models domain-specific notion of "importance" produces good-quality summaries as compared to several baselines. However, there is a large gap between the performance of CONTRASUM and the upperbound, leaving scope for future work including the generation of abstractive summaries.

## 10 Limitations and Future Work

We note the following limitations of this work: (1) We take a non-trivial step of generating partyspecific extractive summaries of a contract to ease contract reviewing and compliance. However, a simplified abstractive summary of key points will be valuable in improving the understanding of a contract. We leave further simplification of extractive summaries to future work due to the unavailability and difficulty in collecting abstractive summaries of long contracts and issues of hallucination and factual errors associated with the existing summarization and simplification systems (Garimella et al., 2022). (2) CONTRASUM may not be able to include sentences (containing obligations, entitlements, or prohibitions) that do not explicitly mention either of the parties (*e.g.*, (a)"On termination of the lease, the apartment has to be handed over freshly renovated." and (b)"The tenant can vacate the place at any point in time without any consequences.") in the extracted summaries. While these sentences contain important obligations for tenant and information that is important for landlord to know, we were constrained by the cases that the prior work (Sancheti et al., 2022) covers as our dataset is built atop that dataset. A more sophisticated system could handle example (a) by identifying the tenant as an implicit participant in the "handing over" event, (and the landlord in example (b) even arguably plays an oblique role as the owner of "the place"); the omission of these cases is arguably due to our method rather than fundamental limitations of the task formulation. Follow-up work could explore these aspects either through the identification of implicit participants, or the expansion of categories. (3) Our data collection and model evaluation is limited to lease agreements; studying the generalization of Importance Ranker and CONTRASUM to other types of contracts can be valuable follow-up work. (4) We believe that the most linguistic and semantic attributes necessary for category classification and importance ranking are captured at the sentence level. Therefore, to improve the heterogeneity of data under resource-limited scenario in-terms of number of contracts that we could annotate, we split each contract at a sentence level. However, it is possible that splitting at sentence level may result in an out-of-context annotation. Due to the complexity of involving the whole context and the cognitively challenging nature of the task, we leave the study of document-level generalizability of the annotations for future work. (5) As a sentence may contain multiple categories, same sentence can be a part of summaries for different categories. We leave further segregation of categories within a sentence and explicit modeling of diversity for future.

## 11 Ethical Considerations

We are committed to ethical practices and protecting the anonymity and privacy of the annotators who have contributed. We paid annotators at an hourly rate of >12.5 USD for their annotations.

**Societal Impact** Advances in ML contract understanding and review, including agreement summarization, can reduce the costs of and increase the availability of legal services to small businesses and individuals. We believe that legal professionals would likely benefit from having auxiliary analysis provided by ML models in the coming years. However, we recognize and acknowledge that our work carries a possibility of misuse including malicious adulteration of summaries generated by our model and adversarial use of Categorizer and Ranker to mislead users. Such kind of misuse is common to any predictive model therefore, we strongly recommend coupling any such technology with external expert validation. The purpose of this work is to provide aid to legal professionals or laypersons dealing with legal contracts for a better understanding of them, and not to replace any experts. As contracts are long documents, a party-specific summary of key *obligations*, *entitlements*, and *prohibitions* can help significantly reduce the time spent on reading and understanding the contracts.

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

## A  Appendix

### A.1  More details on Importance Dataset

We hire 3 lawyers from Upwork (1 female, 2 males) based in India. We instructed the annotators to rate the importance of sentences with the vision of important sentences being part of a summary that can be used for contract review and complaince purposes. We also mentioned to them that these annotations will be used to train a machine learning model to produce a summary of a given contract with respect to each contracting party. We also mentioned that their annotations will be shared anonymously for research purposes. We observed an increase in annotation reliability with the increase in the number of annotations done by the annotators. As the task is subjective, we also observed that the perception of importance changes with years of experience.

**Challenges faced during data collection.** We ran below pilot studies with different annotation task designs to obtain reliable annotations.

- Rating scale importance for each sentence: We provided sentences from a contract to the annotators and asked them to rate the importance level of each sentence with respect to a given party on a scale of 0-5 where 0 denotes 'not at all important', 1 least important, and 5 most important. We asked the annotators to rate sentences from the preamble as least important as our focus is on a scenario where a contract is already been signed. However, we faced inter- and intra-rating consistency issues with this task design.

- Rating scale importance for a pair of sentences: We provided a pair of sentences and a party to the annotator and asked them to rate each sentence's importance level. Providing a pair of sentences provides information on the relative importance. However, we observed inconsistencies in terms of the same sentence being rated with different scores ($\geq \pm 2$).

**Combining annotations.** For combining the annotations, we also experimented with a simple counting-based method (Orme, 2009; Flynn and Marley, 2014); the fraction of times a sentence was chosen as the best (i.e., most important) minus the fraction of times the item was chosen as the worst (i.e., least important). However, there were many ties in the resulting scores as well as the

| Party | Obligation | Entitlement | Prohibition |
|---|---|---|---|
| Tenant | $68.61 \pm 39.39/144$ | $24.23 \pm 20.87/64$ | $8.08 \pm 6.10/24$ |
| Landlord | $23.85 \pm 17.66/63$ | $65.00 \pm 37.88/121$ | $5.30 \pm 5.98/21$ |

Table 6: Average($\pm$ Std)/Maximum number of sentences per contract for each party.

counting-based method did not consider any transitivity relations. Therefore, we used Bradely-Terry model to get real-valued scores from the comparison annotations.

Party-wise statistics of the frequency of sentences in each category per contract are presented in Table 6. As we focus on lease agreements, there are more obligations or prohibitions and fewer entitlements for tenant as compared to the landlord.

### A.2  More details on LEXDEMOD

We present the dataset statistics in Table 10.

### A.3  Evaluation Metrics

The formula for each of the automatic measures is given below. We compute these scores for each category, for each party of a contract.

$$Precision@k = \frac{\text{true positive@k}}{k}$$

$$Recall@k = \frac{\text{true positive@k}}{\text{true positive@k} + \text{false negative@k}}$$

$$F1@k = \frac{2 * \text{Precision@k} * \text{Recall@k}}{(\text{Precision@k} + \text{Recall@k})}$$

$$MAP = \frac{1}{Q} \sum_{q=1}^{Q} \sum_{k=1}^{n} P@k * (R@k - R@(k-1))$$

$$DCG = \sum_{i=1}^{n} \frac{rel_i}{log_2(i+1)}$$

$$IDCG = \sum_{i=1}^{n} \frac{rel_i^{true}}{log_2(i+1)}$$

$$nDCG = \frac{DCG}{IDCG}$$

where $rel_i$ is the relevance (1 or 0) for the predicted sentences and $rel_i^{true}$ is the relevance of the ideal ordering of sentences. We take $n = min(1, \text{number of predicted sentences})$ for NDCG. $Q$ is the number of parties or contracts on which the measure is averaged.

The average score is computed by using the following formula for each metric $m$, $N$ contracts in

| Model | Accuracy | Precision | Recall | F1 |
|---|---|---|---|---|
| Majority | 39.53/28.66/34.38 | 6.49/5.23/11.72 | 14.29/14.29/21.09 | 8.92/7.66/15.03 |
| Rule-based | 61.32/50.54/56.22 | **81.81**/75.21/**80.04** | 46.66/45.54/46.64 | 50.13/46.16/48.76 |
| BERT-BU | 74.07/70.79/72.52 | 73.68/74.44/75.48 | 75.84/71.02/77.17 | **78.81**/71.18/75.61 |
| RoBERTa-B | 75.53/71.42/73.59 | 73.54/72.17/74.48 | 78.39/72.88/78.31 | 74.90/71.91/75.66 |
| C-BERT-BU | 77.50/73.25/75.48 | 76.63/76.22/77.52 | **80.47**/71.54/78.81 | 77.95/72.34/77.67 |
| RoBERTa-L | **78.28**/**75.03**/**76.74** | 75.05/**77.69**/77.30 | 79.59/**75.21**/**79.11** | 76.71/**76.00**/**77.88** |

Table 7: Evaluation results for the Content Categorizer. Scores averaged across 3 different seeds are reported for **Tenant/Landlord/Both** from Sancheti et al. (2022). BU, B, and L denote base-uncased, base, and large, respectively.

| Module | Accuracy | Precision | Recall | F1 |
|---|---|---|---|---|
| Content Categorizer-2 | 76.92 | 77.80 | 77.19 | 77.15 |
| Content Categorizer-3 | 76.29 | 78.89 | 79.50 | 79.12 |
| Importance Ranker-2 | 68.74 | 68.34 | 68.39 | 68.36 |
| Importance Ranker-3 | 69.42 | 69.23 | 69.31 | 69.26 |

Table 8: Evaluation results of content categorizer (-fold) and importance ranker (-fold) for remaining two folds. RoBERTa-L is fine-tuned for both the modules as it was the best-performing model for fold-1.

| | Model | CR=0.05 ROUGE-1/2/L | CR=0.10 ROUGE-1/2/L | CR=0.15 ROUGE-1/2/L |
|---|---|---|---|---|
| PC | Random | 32.51/18.21/21.06 | 41.22/23.75/25.03 | 47.26/28.23/28.39 |
| | KL-Sum | 32.32/17.53/20.71 | 40.26/21.77/23.84 | 46.37/26.21/27.16 |
| | LSA | 32.75/18.05/20.78 | 41.40/23.58/24.86 | 47.16/27.84/27.33 |
| | TextRank | 32.39/18.76/21.55 | 41.73/25.02/25.98 | 48.03/29.33/29.16 |
| | PACSUM (BERT) | **32.85**/18.66/21.35 | 41.75/24.74/25.72 | 48.03/29.63/29.27 |
| | LexRank | 32.78/18.31/21.38 | 42.38/24.67/25.85 | 48.19/28.90/29.32 |
| | CONTRASUM | 32.61/**22.92**/**23.99** | **43.76**/**30.91**/**29.35** | **51.37**/**37.31**/**33.19** |
| PC | Upper-bound | 37.04/30.75/32.04 | 51.39/43.99/44.04 | 59.91/51.50/51.55 |
| GC | Random | 36.40/24.91/26.61 | 45.19/30.54/30.55 | 52.28/36.62/34.38 |
| | KL-Sum | 34.82/21.89/24.27 | 43.86/28.31/38.66 | 50.87/33.76/32.13 |
| | LSA | 36.43/24.24/25.78 | 46.04/30.45/30.51 | 52.77/36.25/33.30 |
| | TextRank | 34.88/23.44/25.32 | 44.56/29.61/29.67 | 51.90/35.93/34.05 |
| | PACSUM (BERT) | 36.19/24.33/26.21 | 45.04/30.10/29.73 | 51.92/36.19/33.64 |
| | LexRank | 35.18/23.13/24.76 | 44.67/29.32/29.40 | 52.24/36.42/35.15 |
| | CONTRASUM-CC | 37.55/32.28/30.84 | 49.16/40.67/36.49 | 57.68/48.37/40.91 |

Table 9: Rouge scores for end-to-end summarization for different compression ratios (CR) averaged across 3 folds. GC (PC) denotes use of ground-truth (predicted from the content categorizer) categories. CC: content categorizer.

| Split | #Sent. | #Spans | Obl | Ent | Pro | Per | Nobl | Nent | None |
|---|---|---|---|---|---|---|---|---|---|
| Train | 4282 | 5279 | 1841 | 1231 | 343 | 289 | 265 | 239 | 1071 |
| Dev | 330 | 421 | 176 | 86 | 20 | 18 | 21 | 22 | 78 |
| Test | 1777 | 1952 | 575 | 418 | 64 | 167 | 101 | 88 | 539 |

Table 10: Dataset Statistics.

the test set with number of parties as $N_{party}$, and $m_k$ metric value for the $k$th category.

$$Avg(m) = \frac{1}{N}\sum_{i=1}^{N}\frac{1}{N_{party}}\sum_{j=1}^{N_{party}}\frac{1}{N_{category}}\sum_{k=1}^{N_{category}} m_k$$

### A.4 Discussion on Choice of $m_i$

Since $m_i$ is capped at 10 for predicted summaries and one predicted summary is compared against references with different compression ratios, there are two cases when a predicted summary can have more sentences (capped at 10) than reference summaries (*e.g.*, number of prohibitions is $< 10$ at $CR = 0.15$ for a party): (1) when categorizer makes false positive predictions (2) when predictions are correct and capped at 10 but the overall number of sentences belonging to a category are less resulting in fewer than 10 sentences in the references for a compression ratio. We believe that the choice of $m_i$ to be kept fixed is justified as there can be false positives or false negatives during category prediction resulting in more or fewer sentences belonging to a category. Also, setting $m_i$ to be the same as reference summaries is not a realistic choice as in a real-world setting reference summaries are not available. However, we also provide results (see Table 12) of the systems when $m_i$ is kept the same as the reference summaries to demonstrate that CONTRASUM outperforms others in this setting as well.

### A.5 Additional Results

**Evaluation results of content categorizer against baselines.** We report the results in Table 7.

**Evaluation results of content categorizer and importance ranker for remaining two folds.** We create three splits of the data for the 3 fold cross-validation of CONTRASUM. Each fold consists of 5 contracts in the test set, 1 in the dev set, and the remaining in the train set. The results presented in the main paper for the content categorizer and the importance ranker are for fold-1 and the results for the other two folds of the dataset are presented in Table 8. As the number of annotated contracts in the LEXDEMOD dataset is more than that of the importance comparison dataset, we always keep the sentences from non-overlapping contracts in the train set for the content categorizer. We run each experiment with 3 seed values (123, 111, 11) and use the predictions from the model which gives the best result on the dev set. The dev set is kept the same across the three folds for both the content

| | Model | Tenant | | | | | | Landlord | | | | | |
|---|---|---|---|---|---|---|---|---|---|---|---|---|---|
| | | CR=0.05 | | CR=0.10 | | CR=0.15 | | CR=0.05 | | CR=0.10 | | CR=0.15 | |
| | | MAP | NDCG | MAP | NDCG | MAP | NDCG | MAP | NDCG | MAP | NDCG | MAP | NDCG |
| PC | Random | 0.102 | 0.162 | 0.116 | 0.195 | 0.139 | 0.228 | 0.124 | 0.176 | 0.130 | 0.203 | 0.144 | 0.228 |
| | KL-Sum | 0.076 | 0.137 | 0.089 | 0.163 | 0.11 | 0.200 | 0.176 | 0.210 | 0.175 | 0.226 | 0.167 | 0.236 |
| | LSA | 0.089 | 0.150 | 0.102 | 0.176 | 0.140 | 0.221 | 0.097 | 0.157 | 0.103 | 0.178 | 0.145 | 0.229 |
| | TextRank | 0.095 | 0.156 | 0.111 | 0.199 | 0.125 | 0.224 | 0.148 | 0.208 | 0.155 | 0.238 | 0.180 | 0.265 |
| | PACSUM (BERT) | 0.151 | 0.208 | 0.167 | 0.257 | 0.168 | 0.268 | 0.119 | 0.180 | 0.129 | 0.206 | 0.133 | 0.229 |
| | LexRank | 0.144 | 0.200 | 0.179 | 0.252 | 0.183 | 0.270 | 0.117 | 0.172 | 0.129 | 0.206 | 0.148 | 0.236 |
| | CONTRASUM | **0.207** | **0.317** | **0.209** | **0.340** | **0.236** | **0.378** | **0.240** | **0.321** | **0.258** | **0.361** | **0.289** | **0.406** |
| PC | Upper-bound | 0.579 | 0.630 | 0.614 | 0.688 | 0.613 | 0.694 | 0.579 | 0.626 | 0.600 | 0.671 | 0.615 | 0.702 |
| GC | Random | 0.195 | 0.298 | 0.196 | 0.313 | 0.221 | 0.349 | 0.217 | 0.298 | 0.221 | 0.319 | 0.236 | 0.343 |
| | KL-Sum | 0.119 | 0.196 | 0.119 | 0.238 | 0.131 | 0.366 | 0.229 | 0.309 | 0.263 | 0.368 | 0.276 | 0.389 |
| | LSA | 0.159 | 0.241 | 0.173 | 0.267 | 0.267 | 0.366 | 0.310 | 0.377 | 0.305 | 0.397 | 0.307 | 0.413 |
| | TextRank | 0.111 | 0.209 | 0.123 | 0.242 | 0.146 | 0.275 | 0.176 | 0.265 | 0.180 | 0.285 | 0.210 | 0.330 |
| | PACSUM (BERT) | 0.142 | 0.256 | 0.161 | 0.289 | 0.192 | 0.321 | 0.182 | 0.253 | 0.194 | 0.293 | 0.228 | 0.338 |
| | LexRank | 0.217 | 0.319 | 0.238 | 0.347 | 0.232 | 0.354 | 0.181 | 0.238 | 0.200 | 0.288 | 0.241 | 0.352 |
| | CONTRASUM-CC | 0.355 | 0.499 | 0.342 | 0.495 | 0.395 | 0.549 | 0.441 | 0.551 | 0.444 | 0.574 | 0.467 | 0.601 |

Table 11: Evaluation results of end-to-end summarization with respect to Tenant and Landlord for different compression ratios (CR) averaged across 3 folds. GC (PC) denotes use of ground-truth (predicted from the content categorizer) categories; CC: content categorizer.

| | Model | CR=0.05 | | CR=0.10 | | CR=0.15 | |
|---|---|---|---|---|---|---|---|
| | | MAP | NDCG | MAP | NDCG | MAP | NDCG |
| PC | Random | 0.042 | 0.052 | 0.054 | 0.080 | 0.074 | 0.118 |
| | KL-Sum | 0.077 | 0.088 | 0.082 | 0.107 | 0.083 | 0.124 |
| | LSA | 0.021 | 0.030 | 0.033 | 0.058 | 0.076 | 0.119 |
| | TextRank | 0.037 | 0.048 | 0.050 | 0.081 | 0.088 | 0.137 |
| | PACSUM (BERT) | 0.053 | 0.064 | 0.065 | 0.100 | 0.079 | 0.138 |
| | LexRank | 0.065 | 0.076 | 0.089 | 0.125 | 0.106 | 0.159 |
| | CONTRASUM | **0.136** | **0.181** | **0.152** | **0.219** | **0.189** | **0.282** |
| PC | Upper-bound | 0.540 | 0.586 | 0.573 | 0.642 | 0.589 | 0.671 |
| GC | Random | 0.107 | 0.126 | 0.117 | 0.157 | 0.132 | 0.193 |
| | KL-Sum | 0.072 | 0.078 | 0.100 | 0.145 | 0.109 | 0.165 |
| | LSA | 0.151 | 0.163 | 0.168 | 0.202 | 0.229 | 0.291 |
| | TextRank | 0.037 | 0.054 | 0.042 | 0.077 | 0.070 | 0.128 |
| | PACSUM (BERT) | 0.056 | 0.067 | 0.069 | 0.104 | 0.107 | 0.169 |
| | LexRank | 0.089 | 0.102 | 0.117 | 0.154 | 0.150 | 0.212 |
| | CONTRASUM-CC | 0.248 | 0.275 | 0.360 | 0.535 | 0.341 | 0.456 |

Table 12: Results of end-to-end summarization for different compression ratios (CR) averaged across 3 folds when the maximum number of sentences per category in the output summary is set same as the reference summaries. GC (PC) denotes use of ground-truth (predicted from CC) categories. CC: Content Categorizer.

categorizer and the importance ranker.

**ROUGE scores for end-to-end summarization task.** We report ROUGE-1/2/L for the summarization task in Table 9. We observe similar results as in Table 4. CONTRASUM outperforms all other baselines on ROUGE-2 and ROUGE-L for all compression ratios. A minor difference in ROUGE-1 scores among different baselines might be because of the limited vocabulary used in legal documents resulting in similar 1-grams. There is a huge gap between CONTRASUM and upper-bound scores showing great room for improvement. This gap increases with the increase in compression ratios.

**End-to-end summarization results for each party.** We present the summarization results with respect to each party averaged over the 3 folds in Table 11 for Tenant and Landlord. We club "tenant", and "lessee" under Tenant, and "landlord" and "lessor" under Landlord for presenting the results. We observe similar trends in the performance of different models as the overall performance presented in the main paper. CONTRASUM outperforms each of the baselines across both the measures and for both the parties. As the number of obligations and entitlements is more than prohibitions and since we cap the maximum number of predictions at 10, as the compression ratio increases the % improvement will be small.

**Qualitative outputs for the end-to-end summarization task from ChatGPT** Since contracts are much longer than the token limit of chatGPT (3.5-turbo), we cannot prompt it to generate the summary of all the obligations for a tenant in one go. Instead, we segment the contract at a page-level and first prompt (*Can you extract sentences that mention any obligations for the Tenant from this text. Ensure that sentences are present in the provided contract.*) ChatGPT to extract all the obligations mentioned for the tenant in the provided text from this contract[8]. Since the output of ChatGPT is not deterministic, after running the same prompt for 5 times, we observed that the output sentences were extractive only 50% of times. Also, generated obligations were sometimes hallucinated; generic but not present in the text of the page provided as context. For *e.g.*, it generated "Section 13.1 of the lease mentions that it is the Tenant's responsibility to maintain the Leased Premises in good condition and repair." as an obligation for the first page in the contract.

---

[8] We experiment for one contract https://www.sec.gov/Archives/edgar/data/1677576/000114420419013746/tv516010_ex10-1.htm

**Full sample summary.** We provide the full sample summary in Figure 4 and Figure 5.

## A.6 Implementation Details

We use Adam optimizer with a linear scheduler for learning rate having an initial learning rate of $2e^{-5}$, and warm-up ratio set at $0.05$. All the models are trained and tested on NVIDIA Tesla V100 SXM2 16GB 904 GPU machine. We experiment with batch size $\in \{2, 4, 8\}$ ($\in \{8, 16, 32\}$), number of epochs $\in \{3, 5, 10, 20, 30\}$ ($\in \{1, 3\}$), learning rate $\in \{1e^{-5}, 2e^{-5}, 3e^{-5}, 5e^{-5}\}$ ($\in \{1e^{-5}, 2e^{-5}, 3e^{-5}, 5e^{-5}\}$), and warm-up ratio $\in \{0.05, 0.10\}$ ($0.05$) for the content selector (importance predictor). BERT-base ($110M$ parameters) and Roberta-base ($125M$ parameters) models took $46$ ($120$) mins, and RoBERTa-large ($355M$ parameters) took $2$ ($6$) hrs to train for content selector (importance predictor). We use choix python package for Bradley-Terry model's implementation. We use official implementation and released models of PACSUM [9].

## B  Human Evaluation

We perform a human evaluation of 16 summaries, 8 each for 2 contracts; 4 for Tenant and 4 for Landlord. Annotators are provided with a contract and the output summaries to rate them on several criteria. Different criteria are defined as:

1. Informativeness: How informative is the summary for the given party from the review or compliance perspective?
2. Usefulness: How useful is the summary for the given party from the review or compliance perspective?
3. Accuracy of Categorization: How correct is the partitioning of the sentences in each category?
4. Redundancy: How much of the content of the summary is repetitive or about the same topic?
5. Accuracy of importance ranking: How correctly are the sentences within a category ranked?
6. Overall: How good is the overall quality of the whole summary?

The annotators were legal experts from India (1 female, 1 male); they went through the contract, and then scored the summaries. For the correctness of categorization, they also considered if the sentence belongs to a category directly or indirectly with respect to a party. For e.g., "Tenant shall pay the rent to the landlord", represents a direct obligation for the Tenant and an indirect entitlement for the Landlord. So, in case this sentence is present in the summary for both tenant and landlord, then categorization score with respect to tenant will be more than for landlord. Usefulness is scored with respect to a summary that the experts have in mind after reading the contract, whereas informativeness is scored only on the basis of the produced summary. We present the category-wise human evaluation scores in Table 13. CONTRASUM obtains good scores in most of the measures and even better than references in a few cases. However, it is rated low on diversity which is possible because we do not explicitly account for diversity during our modeling.

---

[9] https://github.com/mswellhao/PacSum

[10] The contract is available at https://www.sec.gov/Archives/edgar/data/1677576/000114420419013746/tv516010_ex10-1.htm

[11] The contract is available at https://www.sec.gov/Archives/edgar/data/1677576/000114420419013746/tv516010_ex10-1.htm

| Model | Acc. of Categorization↑ | Informativeness↑ | Redundancy↓ | Usefulness↑ | Acc. of Importance Ranking↑ |
|---|---|---|---|---|---|
| Random (PC) | 3.25/2.75/2.75 | 3.00/2.65/**3.25** | **1.50**/2.37/1.87 | 2.75/2.75/3.12 | 2.25/2.50/2.75 |
| LexRank (PC) | 3.00/3.25/2.87 | 2.75/3.37/2.87 | 2.12/**1.50**/2.25 | 2.50/3.37/2.62 | 2.37/2.87/2.25 |
| CONTRASUM | **3.62**/3.12/**3.00** | 3.37/**3.50**/2.87 | 2.25/1.87/2.00 | **3.37**/3.37/**3.37** | **2.50**/2.12/3.00 |
| Reference | 3.25/**3.62**/2.62 | **3.37**/**3.50**/2.62 | 1.87/2.00/**1.37** | 2.87/**3.50**/2.62 | 2.25/**3.12**/**3.25** |

Table 13: Category-wise (Obligation/Entitlement/Prohibition) results for human evaluation of model summaries. Mean scores of the 2 annotators are averaged for the parties and then averaged across contracts.

---

## Obligations

1. If any installment of Rent due from Tenant is not received by Landlord within three (3) days after the date such payment is due, Tenant shall pay to Landlord (a) an additional sum of ten percent (10%) of the overdue Rent as a late charge plus (b) interest at an annual rate (the "Default Rate") equal to the lesser of (a) fifteen percent (15%) and (b) the highest rate permitted by Applicable Laws.
2. Tenant shall pay to Landlord as Additional Rent all sums so paid or incurred by Landlord, together with interest at the Default Rate, computed from the date such sums were paid or incurred.
3. Tenant shall make all arrangements for and pay for all water, sewer, gas, heat, light, power, telephone service and any other service or utility Tenant requires at the Premises.
4. If Tenant refuses or neglects to repair or maintain (or commence and pursue the process of repairing or maintaining) the Premises as required hereunder to the reasonable satisfaction of Landlord, Landlord, at any time following ten (10) business days from the date on which Landlord shall make written demand on Tenant to affect such repair or maintenance, may, but shall not have the obligation to, make such repair and/or maintenance (without liability to Tenant for any loss or damage which may occur to Tenant 's merchandise, fixtures or other personal property, or to Tenant's business by reason thereof) and upon completion thereof, Tenant shall pay to Landlord as Landlord Additional Rent Landlord's costs for making such repairs, plus interest at the Default Rate from the date of expenditure by Landlord upon demand therefor.
5. Tenant shall indemnify, save, defend (at Landlord's option and with counsel reasonably acceptable to Landlord) and hold the Landlord Indemnitees harmless from and against any Claims arising from any such liens, including any administrative, court or other legal proceedings related to such liens.
6. In the event that any utilities are furnished by Landlord, Tenant shall pay to Landlord the cost thereof as an Operating Expense.
7. Tenant shall, at Tenant's sole cost and expense, provide odor eliminators and other devices (such as filters, air cleaners, scrubbers and whatever other equipment may in Landlord 's judgment be necessary or appropriate from time to time) to abate any odors, fumes or other substances in Tenant's exhaust stream that emanate from Tenant's Premises to a commercially reasonable level consistent with the Permitted Use.
8. To the fullest extent permitted by law, Tenant shall indemnify, protect, defend and hold Landlord (and its employees and agents) harmless from and against any and all claims, costs, expenses, suits, judgments, actions, investigations, proceedings and liabilities arising out of or in connection with Tenant 's obligations under this Section 20, including, without limitation, any acts, omissions or negligence in the making or performance of any such repairs or replacements.
9. The following are conditions precedent to a Transfer or to Landlord considering a request by Tenant to a Transfer : Tenant agrees to indemnify, save, defend (at Landlord's option and with counsel reasonably acceptable to Landlord) and hold the Landlord Indemnitees harmless from and against any and all Claims of any kind or nature, real or alleged, arising from (a) injury to or death of any person or damage to any property occurring within or about the Premises arising directly or indirectly out of the presence at or use or occupancy of the Premises or Project by a Tenant Party, (b) an act or omission on the part of any Tenant Party, (c) a breach or default by Tenant in the performance of any of its obligations hereunder, including with respect to compliance with the obligations of Landlord under the Oil and Gas Lease; or (d) injury to or death of persons or damage to or loss of any property, real or alleged, arising from the serving of any intoxicating substances at the Premises or Project, except to the extent directly caused by Landlord 's gross negligence or willful misconduct.
10. Tenant shall reimburse Landlord for all third-party costs actually incurred by Landlord in connection with any Alterations, including Landlord's third-party costs for plan review, engineering review, coordination, scheduling and supervision thereof.

## Prohibitions

1. Except for odors and fumes that are consistent with the Prior Course of Dealing or that are typical and customary in connection with operations permitted under the Oil and Gas Lease, and, which, in any event, are not otherwise a violation of any Applicable Laws or CC&Rs, Tenant shall not cause or permit (or conduct any activities that would cause) any release of any odors or fumes of any kind from the Premises.
2. Each of Landlord and Tenant shall keep the terms and conditions of this Lease confidential and shall not (a) disclose to any third party any terms or conditions of this Lease or any other Lease-related document (including subleases, assignments, work letters, construction contracts, letters of credit, subordination agreements, non-disturbance agreements, brokerage agreements or estoppels) or (b) provide to any third party an original or copy of this Lease (or any Lease-related document).
3. The following are conditions precedent to a Transfer or to Landlord considering a request by Tenant to a Transfer: Whenever consent or approval of either party is required, that party shall not unreasonably withhold, condition or delay such consent or approval, except as may be expressly set forth to the contrary.

Figure 4: Obligations and permissions of full sample summary produced by CONTRASUM with respect to Tenant for a contact[10]. The sentences appear in decreasing order of importance in each category. Entitlements in Figure 5

Figure 5: Entitlements of full sample summary produced by CONTRASUM with respect to Tenant for a contact[11]. The sentences appear in decreasing order of importance in each category. Obligations and Permissions in Figure 4