# OpenReview forum: "What to Read in a Contract? Party-Specific Summarization of Legal Obligations, Entitlements, and Prohibitions"
_EMNLP/2023/Conference — EMNLP 2023 Main_

### Official Review · Reviewer_EMMa · 2023-08-01

**Soundness:** 3

**Excitement:**

3: Ambivalent: It has merits (e.g., it reports state-of-the-art results, the idea is nice), but there are key weaknesses (e.g., it describes incremental work), and it can significantly benefit from another round of revision. However, I won't object to accepting it if my co-reviewers champion it.

**Missing References:**

*Summary of Terms and Condtions*

In addition to the legal document types mentioned in the paper, automated summaries also exist for terms and conditions

T. Perera and T. Perera, "Barrister-Processing and Summarization of Terms & Conditions / Privacy Policies," 2021 6th International Conference for Convergence in Technology (I2CT), Maharashtra, India, 2021, pp. 1-7, doi: 10.1109/I2CT51068.2021.9418090.

Daniel Braun, Elena Scepankova, Patrick Holl, and Florian Matthes. 2017. SaToS: Assessing and Summarising Terms of Services from German Webshops. In Proceedings of the 10th International Conference on Natural Language Generation, pages 223–227, Santiago de Compostela, Spain. Association for Computational Linguistics.

*Question answering *

There is a number of works on question answering on contracts (mostly privacy policies), which are also relevant in the context of personalised summaries.

Wasi Ahmad, Jianfeng Chi, Yuan Tian, and Kai-Wei Chang. 2020. PolicyQA: A Reading Comprehension Dataset for Privacy Policies. In Findings of the Association for Computational Linguistics: EMNLP 2020, pages 743–749, Online. Association for Computational Linguistics.

**Paper Topic And Main Contributions:**

The paper presents an approach to generate extractive summaries of the obligations, entitlements, and prohibitions in tenancy agreements separately for tenants and landlords (i.e. the contracting parties), based on a two-staged approach of a content categoriser and an importance ranker. Additionally, the paper provides a new, annotated, data set for summary generation.

**Questions For The Authors:**

A: Related to my comment above, what is the use case for / value of a summary that only considers obligations, entitlements, and prohibitions of one party?

**Reasons To Accept:**

* The paper presents an interesting application case for tailored summaries
* A data set annotated by legal experts in a domain where very little annotated data exists so far will be published alongside the paper
* The authors conducted a human evaluation of their system

**Reasons To Reject:**

The main limitation of the paper in my view is that it has no notion or discussion of what is important to the parties, which should be the basis for a party-specific summarization. The approach only considers sentences that explicitly mention one of the parties. That introduces two very important limitations that are not addressed in the paper: (1) If a sentence does not explicitly mention either of the parties, it is not considered. The sentence "On termination of the lease, the apartment has to be handed over freshly renovated." clearly contains an (arguably important) obligation of the tenant, but will be ignored in the summary. (2) If the agreement would contain the sentence (excuse the slight exaggeration) "The tenant can burn the place down at any point in time without any consequences." This would not show up in the landlord summary. For me, this raises the question: In which constellation would it be relevant to have a party-specific summary in the way "party-specific" is used in the paper? As a landlord I would certainly want to see that information in the summary, as a lawyer reviewing the agreement for a landlord, I would also certainly want to see that. While a party-specific summary that is tailored to the information needs of one party seems like a good idea to me, it is hard for me to imagine the value of a party-specific summary that only contains the obligations, entitlements, and prohibitions of one party.

Another aspect of importance that is not investigated or discussed by the paper: Is legal importance (which is what the experts annotate) equal to importance for a summary? The authors use the example of "paying rent on time" as an important obligation of tenants. From a legal perspective that's agreeable. For a summary, where brevity is important, it is questionable if this information carries any value to the recipient of the summary because that is information that is probably to be found in every rental agreement and therefore its absence would be more noteworthy than its presence. Some of the existing work suggests e.g. lawfulness (Braun et al., 2017) or risk (Keymanesh et al., 2020) as criteria for importance in a summary. A discussion of the different aspects of importance would have significantly enriched the paper.

**Reproducibility:**

5: Could easily reproduce the results.

**Reviewer Confidence:**

5: Positive that my evaluation is correct. I read the paper very carefully and I am very familiar with related work.

---

> ### Author Rebuttal · Authors · 2023-08-29
>
> We thank the reviewer for constructive feedback and great points to add as discussion. Thanks for pointing out the missing references, we will add them in the camera-ready.
>
> - During annotation, the party perspective is not taken into account: We would like to clarify that we did consider party-specificity during importance annotation (L182-185) as well as modeling of the Importance Ranker (L384). The experts were asked to choose the most and least important sentences (out of 4 sentences) from the perspective of a given party.
>
> - limitations that are not addressed in the paper: We agree that such cases may not be covered by our system. However, we were constrained by the cases that the prior work (Sancheti et. al, 2022) covers as we build our dataset atop the dataset introduced in that work. Including such cases during the importance annotations alone would not have helped as our categorizer is developed using prior dataset; such sentences may be categorized as none and thus may not become part of the summary since ContaSum is a pipeline-based system. In principle, a more sophisticated system could handle example (1) by identifying the tenant as an implicit participant in the “handing over” event,  (and the landlord in example 2 even arguably plays an oblique role as the owner of “the place”); the omission of these cases in which parties play an implicit or oblique role is arguably due to our method rather than fundamental limitations of the task formulation. However, this is an important point which could be explored further in follow-up work, either through the identification of implicit participants, or the expansion of categories. We will clarify this in the limitations section. We would also like to point out that the deontic modality dataset does cover instances such as  “Tenant agrees to indemnify the Landlord from all costs and charges for the Utility Services consumed on the Leased Premises” wherein the obligation of the tenant is also considered an entitlement for the Landlord and therefore can be included in Landlord’s entitlement summary. However, we agree that in case Landlord is not mentioned in the sentence, it may not be included in the summary as described above with the current methods, and this is a noteworthy limitation.
>
> - Is legal importance (which is what the experts annotate) equal to importance for a summary?: This is an excellent point. In our annotation process, annotators were made aware of the end-purpose of their annotations (L200-203), so we believe that the annotators were sensitive to factors like “paying the rent on time” being expected/unsurprising. We did learn from interaction and feedback with our annotators about several factors they considered when determining importance. These factors included the degree of liability involved (e.g., ‘blanket indemnification’ were scored higher than ‘costs incurred in alterations’ by a tenant since blanket rights can have an uncapped amount of liability), and liabilities incurred by a more probable event were rated as more important than those caused by less probable events, among others. As is evident from these examples, the factors that can influence importance may be complex, multi-faceted, and difficult to exhaustively identify, which is why our approach in this work was to allow our annotators to use their legal knowledge to inform a holistic judgment. However, we fully agree that a discussion of the factors involved here is well warranted and we will include a discussion of these factors based on annotator feedback in the final camera-ready.

---

### Official Review · Reviewer_CFoT · 2023-08-07

**Soundness:** 4

**Excitement:**

4: Strong: This paper deepens the understanding of some phenomenon or lowers the barriers to an existing research direction.

**Paper Topic And Main Contributions:**

Summary: This paper investigates how summarization of legal contracts can be improved by providing individual summaries to each party involved in a contract. Towards this goal, they introduce a dataset and model for identifying important sentences in lease agreements, focusing on rights and obligations of each party in the contract.

Key contributions:

1. A dataset of 11 contracts (3000 sentences) annotated by experts for party-wise importance.
2. An extractive summarization model that identifies sentences describing prohibitions, entitlements, and obligations, ranks them by importance and generates party-specific summaries
3. Empirical and human evaluations demonstrating the value of the system.

**Reasons To Accept:**

I believe this paper introduces an important task, and the introduction does a good job of describing the advantages of a system that can summarize contracts party-wise. The dataset that will be released by the authors seems like a great resource not just for improving legal document summarization, but summarization systems overall, since datasets that focus on this type of entity or aspect based summarizations are rare. I also think the annotation process described here is interesting, since it doesn't require annotation guidelines specific to the domain, that lay out what makes a sentence important. However, the fact that this information is implicit in the rankings provided by the legal experts makes the data and corresponding systems slightly opaque -- I believe it might be useful to see more analysis or interpretation of these scores apart from what is described in Table 2, such as qualitative comments by the annotators on why a certain decision was made, or cluster analysis for high-ranking or poorly ranking sentences. Apart from this, the dataset curation process seems very sound, and the methods described here could influence the development of further summarization datasets.

**Reasons To Reject:**

I do not think this paper should be rejected, but I am not sure if the chosen baselines are strong enough to illustrate the strength of the system. All the baselines are unsupervised, and the only system (apart from the proposed model) that makes use of the annotated data is the upper bound. I would be curious to see how the contra-sum model performs against a supervised system such as in [1]. I'm also surprised by the human evaluation results that rate the system outputs higher than the gold reference for multiple aspects. While the authors correlate this to the size of the summary, I believe at least for the purposes of the human evaluation that the system output length should have been made comparable to the reference.
[1] Shashi Narayan, Shay B. Cohen, and Mirella Lapata. 2018b. Ranking sentences for extractive summarization with reinforcement learning. In Proceedings of the 2018 Conference of the North American Chapter of the Association for Computational Linguistics: Human Language Technologies, Volume 1 (Long Papers), pages 1747–1759, New Orleans, Louisiana

**Reproducibility:**

2: Would be hard pressed to reproduce the results. The contribution depends on data that are simply not available outside the author's institution or consortium; not enough details are provided.

**Reviewer Confidence:**

2: Willing to defend my evaluation, but it is fairly likely that I missed some details, didn't understand some central points, or can't be sure about the novelty of the work.

**Typos Grammar Style And Presentation Improvements:**

The paper was well-written overall, but I believe Section 4, and in particular 4.2 could be re-written with illustrative examples to make the process clearer. The current text also repeats content from the introduction and Section 3, which could be removed. Please use algebraic notations consistently: such as using a variable for party and categories as well and mathematically describing the classifier's objective, or only use plaintext descriptions throughout and avoid the Ci, Cj notation.

Typos: Bradley misspelled in line 315.

---

> ### Author Rebuttal · Authors · 2023-08-29
>
> We thank the reviewer for the constructive feedback and helpful suggestions. We will incorporate the writing and presentation-related suggestions in the camera-ready.
>
> - Comparison against a supervised baseline: We chose not to train a supervised system because it requires a large amount of annotated (document, summary) pairs. We had 11 document-summary pairs at a contract-level which is not enough for training an end-to-end supervised system such as in [1]. Furthermore, obtaining a large number of summary annotations at a contract-level will be expensive as well as cognitively demanding. This is one of the benefits (L107-110) of our approach which breaks the complex task of party-specific category-wise summarization into sub-tasks (multi-label classification, and pairwise comparison at sentence-level) for which annotations are collected at a sentence or sentence-level pair at a large-scale with lesser cognitive demand. This enables us to use existing 5K sentence-level annotations at a sentence-level for the categorizer and collect 148K labeled sentence-pair level samples (from the 11 contracts) that can be used for training the importance ranker (as opposed to 11 contract-summary pairs to do an end-to-end summary generation). Furthermore, summaries are generated specific to each party in our case. Therefore, it is not straightforward to use [1] as a baseline for these two reasons.
>
> - Surprising human evaluation results: For both the predictive and reference summaries, we established a maximum number of sentences allowed in each category (10 sentences). This maximum was the same for both model output and reference summaries. However, because the category labels themselves were predicted, the system and reference summaries did not always contain the same number of sentences per category if one was below the allowed limit (and in principle it is not possible to enforce a minimum number of sentences per category). One way to control for this length mismatch would have been to perform the human evaluation instead on system outputs conditioned on gold categorizations; however, because we did not have the resources to perform a human evaluation for all settings, we instead reserved the human evaluation for comparing the full end-to-end system outputs against the reference summaries. We agree it is suprising that possible miscategorizations led to higher human ratings of overall summary outputs; this is an unexpected finding that highlights the potential challenges of evaluating complex, structured summarization outputs such as in this task. We will include a discussion of this challenge and possible approaches for follow-up work in the final version. We also provide details on what rationales experts used during the evaluation in L1148-1168.
>
> - We will add qualitative comments by the annotators on why a certain decision was made in the camera-ready. The different factors used by the annotators include the degree of liability involved (e.g., ‘blanket indemnification’ was scored higher than ‘costs incurred in alterations’ by a tenant since blanket rights can have an uncapped amount of liability), and liabilities incurred by a more probable event were rated as more important than those caused by less probable events, among others.
>
> - Reproducibility: We have submitted the dataset along with this submission and will publicly release it upon acceptance. The contracts used are publicly available and the LexDeMod dataset can be accessed by directly contacting the authors.

---

### Official Review · Reviewer_at5K · 2023-08-12

**Soundness:** 4

**Excitement:**

4: Strong: This paper deepens the understanding of some phenomenon or lowers the barriers to an existing research direction.

**Missing References:**

[Wilson] Wilson et al. The Creation and Analysis of a Website Privacy Policy Corpus. ACL 2016. https://aclanthology.org/P16-1126/

...has also curated a dataset based on legal expert annotations.

[Fok] Fok et al. Scim: Intelligent Skimming Support for Scientific Papers. ACM IUI 2023. https://dl.acm.org/doi/abs/10.1145/3581641.3584034

...presents an extractive summarization system for helping humans process academic documents using a slightly different taxonomy but a similar overall approach.

**Paper Topic And Main Contributions:**

The paper describes a party-specific legal document summarization system, moving from a newly-annotated dataset to a system that seems useful to help understand leases.

Overall, I found this paper to be pretty strong, with my main concern being regarding the reliability of the annotation process.

**Questions For The Authors:**

There are some questions in my review, the answers will not particularly impact my scoring of the paper. I mostly pose questions in reviews to expose areas I found confusing as a reader or help prompt others' thinking.

**Reasons To Accept:**

Overall, I am firmly on the fence for this paper. Overall I liked it and found it informative, but am not sure if the underlying annotation reliability issue is a showstopper at this time.

- Problem is well chosen, but motivation has room for improvement

Legal documents touch a wide variety of people, many of whom are ill-prepared to understand and execute contracts correctly. Improvements in allowing people to do so has obvious societal benefit

- Paper describes the pipeline well

I think I could work with a student and stand up something similar

**Reasons To Reject:**

Overall, while I am on the fence and this section is longer than the "reasons to accept," I think all but the first concern (annotation reliability) are fixable within this round of review.

- Not clear that the data annotations are reliable

L142-163 describe how the dataset is composed of US contracts, but Appendix A.1 mentions that the lawyers annotating these contracts are based in India. While this experimental design decision makes sense as a cost-control mechanism, there are substantial differences in the legal systems of these two jurisdictions. This means it bears some discussion about how this potential mismatch between annotator and data corpus might affect the labelling and downstream systems that learn from this labelling. Similarly, it isn't clear that this data set size is sufficient to train a high-capacity model.

- Not clear that (or how) this extends to more complicated legal frameworks.

The paper restricts itself to lease agreements, which is fine (around L90). However, the paper should contain some discussion about generalization questions beyond what we find around L680 in the limitations section. As I read the paper, I was brainstorming a bit about examples beyond "employment agreements," because those share the 2-party asymmetric role structure of a lease agreement. In particular, I thought about multilateral trade agreements, which are multi-party and receive far more scrutiny than housing rental contracts due to the higher stakes. As is, I don't envision many prospective renters or landlords using such a system beyond curiosity. While I recognize that expanding the universe of contracts studied will be laborious, this concern can be dealt with more cheaply by painting a vision early in the paper of the future where diverse types of legal documents have tools to assist laypersons parse said documents, then present this paper as a necessary step along that path.

- Expand on several concepts...

1. ContraSUM methodology: Most of the novelty in this paper lies in the methodology to handle 2-party contracts with content categorization and so on. Parts of the document focusing on results that are peculiar to housing leases are less interesting and are candidates to shorten (e.g., around L240).
2. Why the BWS scheme described around line 180 is important: I recognize the savings that comes from providing a partial ordering in this way, but it is not clear why this ordering is important in the first place. It seems like Appendix figures 4 and 5 list the sentences in an order based on this importance scores. However, it does not appear to be used for filtration or anything further. Given the design decision that the user will observe all the prohibitions/entitlements/obligations anyway, it seems plausible just list the detected sentences in the order that they appear in the document. In so doing, the labelling effort would be substantially lighter.
3. Content categorizer baselines: Providing 1-2 sentences saying what the model is and by what measure the prior work deemed it "best performing" around L446 would be helpful to a reader.
4. Footnote 3: It isn't clear to me as a reader what the conceptual separation between entitlement and permission categories are. This whole section would benefit from greater clarity on the definition of all of these categories, as they are critical to understanding the paper.

**Reproducibility:**

4: Could mostly reproduce the results, but there may be some variation because of sample variance or minor variations in their interpretation of the protocol or method.

**Reviewer Confidence:**

3: Pretty sure, but there's a chance I missed something. Although I have a good feel for this area in general, I did not carefully check the paper's details, e.g., the math, experimental design, or novelty.

**Typos Grammar Style And Presentation Improvements:**

- The bit about LLMs around L560 felt a bit hollow

As written, the paper seems to insinuate that ChatGPT DID produce output on this problem (or perhaps output on the meta-problem of being asked if it is capable of doing this problem), and that this answer indicated unsatisfactory performance for various reasons. If that is the case, the paper should say more about what the "hallucinations" looked like and so on. If that is not the case, the paper should clarify the non-pursuit of that line of inquiry for the provided reasons, as opposed to the pursuit of that line of inquiry and discovery that it was not promising.

- Consider reorganizing Introduction and related work.

While I know placing related work before conclusion is conventional by some standards, it is not my preferred formatting. In this case, it has caused background to blend into both introduction and related work. Further, the last line of section 8 is moving into methodology, suggesting that the paragraph contains a nice segue from this section into what is currently section 4.

---

> ### Author Rebuttal · Authors · 2023-08-29
>
> Thank you for your encouraging feedback and constructive suggestions. We will incorporate the writing, presentation, and missing references-related suggestions in the camera-ready.
>
> - Not clear that the data is reliable: We screened annotators based on their prior experience with US contract drafting and reviewing (annotators had worked with US-based clients). We agree that importance ranking is a complex task and there will be some level of subjectivity. However, we observed good annotation reliability scores (SHR=0.66; L236-237) with best-worst scaling annotation design. As mentioned in L400-413, the annotated dataset for the two sub-tasks at a sentence (5K) and sentence-pair (148k) level are large enough to train high capacity models via fine-tuning. However, training a supervised system for generating category-wise summaries would require a large amount of agreement-summary paired dataset which is quite challenging and expensive to obtain as legal experts are required. This is one of the benefits of our approach which breaks down the difficult task of document-level end-to-end summarization into two sentence-level sub-tasks.
>
> - Not clear that (or how) this extends to more complicated legal frameworks: This is a good point, we will add the below discussion in the camera-ready. Our annotation framework could be extended to include each pair of contracting parties in a multi-party contract. As one party can have obligations towards any of the other parties, Best-worst scaling could be done per pair of parties. For  e.g., the annotation task could be defined to find the most and least important obligation of Party A towards party B. Similarly from modeling perspective, both parties could be specified in the input.
>
> - Expand on several concepts:
> 1. We agree that the lease-specific findings were not interesting but we mentioned it to reinforce the properties of the annotated dataset. We will shorten the findings as suggested.
> 2. As mentioned in L167-178, we ran multiple pilot studies and obtained low agreement when sentences were rated independently, as annotators found the task difficult due to high subjective nature of the task. We found BWS to be the most suitable for this task achieving good annotator reliability scores  (SHR=0.66; L236-237). Detailed discussion about challenges faced during annotation process is provided in A.1.
> 3. This is a good suggestion. We will add that the best-performing model was chosen on the basis of the best F1-score on the dev set and will add a few lines describing the baselines for content categorizer which were different transformer-based models.
> 4. Entitlement is defined as a right to have/do something while permission refers to being allowed to have/do something. We will clarify this difference in the paper.
>
> - ChatGPT : what the hallucinations look like: We have provided a discussion and an example of hallucination in A.5 (L1085-1108) due to limited space. For e.g., ChatGPT generated “Section 13.1 of the 1102 lease mentions that it is the Tenant’s responsibility to maintain the Leased Premises in good condition and repair." as an obligation for the first page in the contract (link in the paper). However, that information was not mentioned in that page.

---

### Meta-Review · Area_Chair_DtPF · 2023-09-23

**Recommendation:** 5

**Metareview:**

The paper introduces a new dataset and pipeline model for the task of party-specific legal document summarization, focused on leases. The goal is to produce summaries for each party containing specific information regarding obligations, entitlements, and prohibitions.

Reviewers agree that this an interesting use case of tailored summaries. They also agree that the description provided for the pipeline apporach is adequate. The data annotation procress is interesting and, while some concerns were raised regarding the suitability of the annotators, these were clarified in the authors' rebuttal. Authors are encouraged to include this additional information in the paper.

Some concerns were raised regarding the suitabilty of the dataset for training models due to its size. The authors' rebuttal clarified that, indeed, the size is a limitation for training end-to-end models, and that's why they proposed a different pipeline-based approach. This is also the reason why no fully-supervised baselines were included in the comparison. Finally, other concerns were raised regarding relevant  statements that could be missing, either due to not explicitly mentioning the party involved, or because "importance" is subjective. Interesting clarifications on these details were provided, as well as acknoledgments of limitations of the paper. Authors are encouraged to include them in the paper.

---

### Decision · Program_Chairs · 2023-10-07

**Decision:**

Accept-Main

**Comment:**

The paper introduces a new dataset and pipeline model for the task of party-specific legal document summarization, focused on leases. The goal is to produce summaries for each party containing specific information regarding obligations, entitlements, and prohibitions.

Reviewers agree that this an interesting use case of tailored summaries. They also agree that the description provided for the pipeline apporach is adequate. The data annotation procress is interesting and, while some concerns were raised regarding the suitability of the annotators, these were clarified in the authors' rebuttal. Authors are encouraged to include this additional information in the paper.

Some concerns were raised regarding the suitabilty of the dataset for training models due to its size. The authors' rebuttal clarified that, indeed, the size is a limitation for training end-to-end models, and that's why they proposed a different pipeline-based approach. This is also the reason why no fully-supervised baselines were included in the comparison. Finally, other concerns were raised regarding relevant  statements that could be missing, either due to not explicitly mentioning the party involved, or because "importance" is subjective. Interesting clarifications on these details were provided, as well as acknoledgments of limitations of the paper. Authors are encouraged to include them in the paper.